# The Role of Economic and Innovation Initiatives in Planning a Smart City Strategy in Greece

Georgios Siokas * and Aggelos Tsakanikas

Laboratory of Industrial and Energy Economics, School of Chemical Engineering,
National Technical University of Athens, Zografou Campus, 9 Iroon Polytechniou Str., 15780 Zografou, Greece;
atsaka@central.ntua.gr
* Correspondence: geosiok@mail.ntua.gr

**Abstract:** As digital technology has become an integral part of urban life's daily operations, the urban landscape is constantly evolving with the needs of its society. This new reality has allowed municipalities to invest in technologies related to smart cities and to exert a greater influence on the national and local economy. In line with this, the paper aims to understand the mechanisms of planning and implementing a municipality's strategy in Greece to exploit the smart city benefits and to foster economic development. It is important to identify the role of different factors including strategy during the planning and implementing phases of initiatives concerning the economy and innovation in a smart city. To achieve this, data were collected via a questionnaire and processed using the advanced statistical technique PLS-SEM. The main findings highlight the importance of planning initiatives aligned with the needs of the municipality and the business ecosystem. The existence of a smart city strategy has a catalytic effect on the final impact of the implemented initiatives on the urban ecosystem. A systematic analysis of the smart cities' dynamics and the new state of the urban ecosystem can help the local actors focus on value creation and public service provision, fostering innovation and profitability.

**Keywords:** smart city ecosystem; urban economy; smart city; entrepreneurship; smart economy

## 1. Introduction

### 1.1. The Need for a Smart Urban Ecosystem

During the last decade, the citizens' needs, challenges, and lifestyles have been diversifying, transforming and evolving the urban ecosystem. Over the decades, the municipal authorities have been faced with increasingly complex problems and challenges, extending to areas such as public transportation and services, health care, the environment, energy, and national, individual, and cyber security. During its transformation phase, resources gradually decreased, the management capacity of municipalities was undermined, and the natural resource management mechanisms and action planning and decision-making were reshaped [1].

Urbanization led progressively to the abandonment of rural areas and the accumulation of large numbers of residents in large urban centres. According to the United Nations' research, currently the urban population exceeds 3 billion, with over 70% of the world's population accumulated in megacities [2] and covering only 2% of the Earth's surface [3]. This has affected the relationship of the citizen to public governance and stresses the value of functions and processes aligned with the daily activities of organizations and inhabitants. Modern urban areas have utilized information and communication technologies (ICT) as the appropriate tools for facing the emerging challenges.

As traditional cities become more intricate, they are transformed into smart cities, confronting the necessities of the political, social, cultural, and financial environment and the capacities of conventional and rigorous administration and infrastructure [4]. While

the urban ecosystem is being "digitalized", as it is composed of multiple smart services [5], the role of the municipality and its local authorities cannot remain unaffected. An urban ecosystem in a smart city is composed of its physical, i.e., infrastructure, and non-physical assets, i.e., public services and history, and of different complex traditional and modern subsystems, i.e., urban transportation and smart sensors.

Advancements and cutting-edge technologies are encountered as a force for change, economically, socially, cultural, and politically affecting all the components of an urban ecosystem. Particularly, ICT ignited a technological revolution, influencing all sectors of the global and urban economy [6], responding to the emerging needs of urban inhabitants and significantly affecting public governance at all levels of policy and management. In numerous instances, the predominant emphasis of the local authorities and researchers has been on potential investments in technological solutions rather than on other issues falling within their responsibilities.

The overall management of the services and the available human and financial capital concern all the sectors at once. Local authorities can gradually support decentralized administration, motivate citizens, and save resources while providing integrated and rapid services. Traditionally, a city's operational model is usually characterized by low connectivity and efficiency and the absence of the ability to carry out horizontal innovation between systems. This model is changing, and the upgraded operating model offers high accessibility and a greater degree of interaction between the different sectors and participants [7]. Thus, the conditions formed lead to the creation of interactive channels, allowing for the entry and promotion of innovation from multiple sources. Innovation has acquired a more substantial and active role, leading to (1) direct and indirect effects on investments and quality of life, (2) the formulation of partnerships, and (3) a variety of roles depending on its goals and strategy. As smart cities continue to develop, the degree of interaction among their various sectors, sources, and citizens intensifies and becomes more sophisticated.

A fifth generation of smart cities is emerging from its aforementioned evolution and trying to improve the aforementioned mismatch. Instead of an approach through technological providers (Smart City 1.0), or through a model driven by the coexistence of technology and people in the city (Smart City 2.0), a model of "co-creation" appears, in which society and citizens are co-creators (Smart City 3.0); or it may be a conceptual model of a complex multi-agent system, a common concept with Industry 4.0 (Smart City 4.0); or a new model where a city is an urban ecosystem of smart services (Smart City 5.0). The policies, regulations, governance, technological agents, and community—citizens and organizations—interact, exchange data, and actively participate to improve urban initiatives and processes on a daily basis [5,8,9]. At the same time, ICT tools, like artificial intelligence, are exploited throughout different smart services to harmonize the co-existence of all different aspects of urban life. Meanwhile, the inhabitants play an essential role in the development and operation of the urban ecosystem while given the possibility to use the open tools and methods for the overall management of the city.

According to our analysis of the literature, an urban ecosystem is distinguished by three main dimensions: (1) the technological, (2) the institutional, and (3) the social human [10]. The impact of ICT in all these dimensions has expanded on six main smart axes: (1) economy, (2) mobility, (3) environment, (4) people, (5) living, and (6) governance [11]. At present, the primary emphasis is placed on the implementation of ICT in various aspects of urban life, with less considerations to other methods such as education and policies that foster innovative technological companies. Specifically, initiatives for a smart economy can be studied at the global, national, regional, or city level. At the urban level, the main structural feature is cities [12] and this is where the role of a smart city is emerging.

*1.2. The Research Axis of This Paper*

Consequently, this paper aims to understand the role of planning a strategy and implementing its methods related to the economy and innovation in a smart city (Figure 1).

It tries to identify the impact of different factors during the design and execution of a strategy. The factors may relate to the needs of the society and the municipality. The primary objective of a city, as posited in the paper, is to reach a state where the city, via its strategy implementation, becomes a fully functional sustainable smart city.

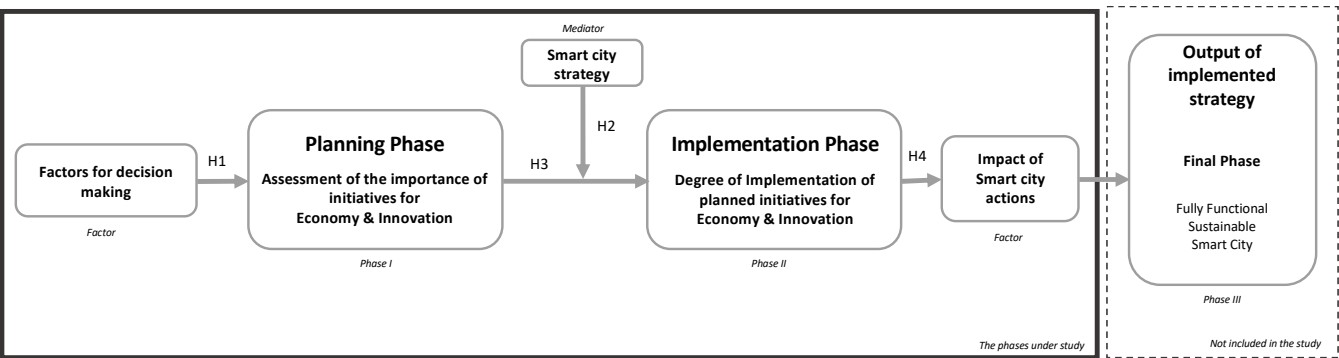

**Figure 1.** The proposed research model integrated with the proposed hypotheses.

There are numerous smart cities in the world that focus on initiatives to further their smart economy. Six cities in Europe (Bristol, Manchester, Barcelona, Amsterdam, Helsinki, and Lisbon), one in North America (New York), one in South America (Medellin (or Medellin)), one in Africa (Johannesburg), and two in Asia (Singapore and Seoul) focus on increasing competitiveness and innovation through their actions. Some examples of these initiatives are that (i) Bristol has developed a platform for advancing technologies and 5G connections, (ii) Manchester has developed the "Smart innovation and People" project, (iii) Amsterdam has the "Smart City Amsterdam" initiative, (iv) Lisbon has Ignite, Helsinki has the "Helsinki Living Lab", and (v) Singapore has the "Smart Nation" initiative.

This paper starts with a summary of the theory and the hypotheses supporting the purpose of this research. Then, the method and the dataset used are presented, followed by the results and their discussion. Finally, the paper concludes with the general output of this research.

## 2. Theoretical Background and Hypotheses Development

The development of the urban ecosystem is accompanied by the upgrading of (1) urban infrastructure, (2) the daily functions of citizens, (3) the development of technologies, and (4) the evolution of civilization [13]. However, for there to be substantial growth, the city needs to implement a sustainable strategy through the development of new technologies [13] and shape a dynamic urban ecosystem, which encourages the collaboration of different organizations, policy makers, local governments, and citizens [14]. The mechanism for creating business opportunities is multidimensional and complex, with continuous interactions, supports the development of a supportive ecosystem, and encourages innovative entrepreneurship [15]. Therefore, there is opportunity for companies and citizens to take risks, innovate, offer advanced solutions that are environmentally friendly, contribute to solving social problems, and help create local employment opportunities [16].

At the same time, the use of modern technologies has enabled the collection and distribution of data and knowledge to provide efficient and sustainable public services [14]. By allowing access to data, local governments encourage the formation of innovative businesses with social and economic value [17]. Given the potential, companies are investing huge resources in smart city initiatives, which promote innovative technologies, support the efficiency of municipalities, and create new business opportunities [18].

### 2.1. Innovation in Smart Cities

Innovation has a catalytic role in the development of a smart city. For an urban ecosystem, innovation is the mechanism for economic development and digital transfor-

mation. Through innovation, local authorities and actors can deal with emerging urban problems [19]. By taking advantage of ICT developments, the implementation of plans, ideas, and visions is achieved, while improving public functions and the quality of life [20].

Beyond the development and use of technology, innovation can revolutionize the applied policy and management practices to align with the city's needs [21]. Through innovation, an organization can adapt to market changes and technological developments, seize emerging opportunities and successfully endure financial problems, i.e., recessions [22]. However, these new conditions call on local authorities to separate themselves from their established practices and promote innovation in public action [23]. Therefore, innovation retains an important role in the economic growth of an urban ecosystem.

## 2.2. The Economy from the Smart City Point of View

An urban ecosystem can act as a driver for the development of innovation. Meanwhile, another economic factor, competitiveness, is inextricably linked to the development of the country and society [24]. Based on the literature, a city's future development is directly linked to its ICT funding. Technologies demonstrate a catalytic impact in leading the digital transformation of the urban ecosystem and economic development on a local, national, and global level. Therefore, the integration of ICT in the economic activities of city's private and public entities transform the urban economy into a "smart economy". This transformation includes factors such as competitiveness, entrepreneurship, brands, innovation, productivity, market flexibility, work, and the integration of the activities of local organizations into the national and global market [25].

The smart economy, as one of the main smart axes of a smart city, is a central element of the economic, social, and cultural development of the urban ecosystem [26]. It is directly linked to national development. For the purposes of this paper, the smart economy treats ICTs as a driver of growth, which contribute to the characteristics of existing and new economic sectors of an urban area and are connected to characteristics relating to (1) an attractive ecosystem for setting up new businesses, (2) creating support mechanisms (i.e., incubators), and an incubator for new businesses, (3) cooperative joint ventures, and (4) collaboration with university research groups. Therefore, the smart city economy is considered "smart" through competition, cooperation, and the clustering of economic units and activities stimulating innovation.

## 2.3. Factors Impacting the Decision-Making Process

As the cities are redefined and new innovations, technological advancements, and smart solutions are invested in for improving the quality of life of inhabitants, the changes are accompanied by new challenges that require new approaches in city management [27] and its decision-making process. Currently, local authorities have at their disposal powerful ICT tools which exploit the power of big data via artificial intelligence, machine learning, and data mining [28,29]. These powerful statistical tools transform the decision-making process. The high-frequency data collected have a high impact in the final decisions made by the responsible parties [30].

The global economy evolves and adapts to technological progress, and the evolution of society and the smart economy is intertwined with the development of ICT and smart cities. During the industrial age, the development of the economy was based more on the linear connections between industries, such as value chains, a company's physical assets, and metrics such as Gross Domestic Product (GDP). On the contrary, the digital age is characterized by networks or sets of sub-networks, fundamental to market rules and mechanisms [31]. Also, it is distinguished by specialization at the individual level, adaptation to mass changes, horizontal exploitation of ICT, the constant exchange of information between different disciplines and actors, and the creation of an ecosystem to support innovative ideas. Through ICT, new businesses are created with (1) new innovative products, (2) the possibility of working from home, (3) smart infrastructure, (4) better access to public documents, (5) improved connectivity of different organizations in public and

private sector, (6) the ability to work faster and more seamlessly, and (7) products/services tailored to their customers' needs [32]. Therefore, local authorities try, via technological developments, to solve critical challenges, to integrate heterogeneous Big Data, strengthen the cognitive level of employees, and increase the possibility of exploiting innovation. All of these allow organizations to solve emerging problems.

Therefore, during the planning of actions related to enhancing the economy and innovation, a municipality needs to focus on the city's needs.

**H1.** *The city's needs impact positively and significantly on the planning phase of a strategy focusing on the smart economy and innovation.*

*2.4. Smart City Strategy*

In the literature, there is a considerable interest in identifying various resources and factors (i.e., approaches, initiatives, projects, and policy aspects), to explore and map the potentials of smart cities across different urban sectors [33]. Cities allocate diverse technological, financial, and human resources by implementing "smart" actions that prioritize the city's requirements, leading to digital transformation [34]. Therefore, integrating technological advancements into policy and decision-making processes at an urban level is crucial [35]. The importance of this integration dwells in the significance of actions (e.g., project, strategy, or isolated actions) that comply with the city's policy and planning, thus emphasizing the significance of customized strategies [36]. Hence, a municipality needs a well-organized smart city strategy oriented on the economy and innovation in order to be able to execute initiatives focusing on creating an attractive ecosystem for setting up new businesses; incubators or other support mechanisms for new businesses; supporting cooperative joint ventures; and helping form urban actors or collaborations with universities and research centres (H2).

**H2.** *The assessment of planning actions in a strategy relating to initiatives focusing on economy and innovation has a significant and positive effect on the degree of implementation of relevant actions and projects.*

Formulating a smart sustainable city strategy requires policymakers to understand and consider the municipality's financial, social, and cultural environment and needs [37]. These strategies should gradually consider the urban area's policy and stakeholder's concerns and the financial, governance, and environmental issues and agendas, including initiatives which support innovation, investment, and partnerships between different sectors and stakeholders [38].

Based on the findings from the literature, municipalities with a well-organized strategy for smart city initiatives have a competitive advantage when carrying out their anticipated initiatives. The impact is noteworthy, particularly when the projects encompass city planning, digital government and services, and educational, financial, and resource management [34]. A coherent and detailed framework can work as a blueprint for city initiators, enabling the effective development of a smart city agenda [39]. The merit of a standardized strategy is observed when significant obstacles to transforming an urban area into a smart one arise in its absence [40]. Therefore, crafting and implementing customized strategies and projects in Greek municipalities can stimulate diverse elements that facilitate and support the smooth transition to digital transformation [34]. An even greater interest is in the correlation of the existence of a structured strategy to the implementation phase of an expected smart city project focusing on the economy and innovation (H3).

**H3.** *The presence of a structured strategy affects indirectly, positively, and significantly the implementation phase of scheduled smart city initiatives related to the economy and innovation.*

*2.5. How Smart Cities Help the Urban Ecosystem*

Cities, as functional systems of physical objects and citizens, consume resources and services and offer economic, social, cultural, and environmental services to satisfy the needs of their inhabitants [41]. The dynamics of cities have greatly permeated all aspects of society and greatly influenced the development of the economy [42]. This aspect of them favours creativity, innovation, and entrepreneurship and highlights the need to develop an entrepreneurial ecosystem, with an emphasis on the economy and entrepreneurial opportunities [43].

The economy, based exclusively on the traditional form of industry, is being transformed into the digital economy, i.e., the smart economy. The smart economy leverages ICT to stimulate innovation through the competition, cooperation, and clustering of economic units and activities. Everyone's role and ability to influence the economy has changed dramatically. End-users have acquired an increased role; their views and society's values influence the decision-making process in both the private and public sectors. The demands of society and the ever-growing needs of cities have created a new reality.

A healthy digital business ecosystem is expected to deliver economic growth, environmental sustainability, and social progress [41]. Subsequently, the city's business ecosystem contributes to growth, sustainability, and quality of life. Meanwhile, the continued upward trajectory of the ICT industry highlights the dynamics of digital transformation and its ability to generate multiple benefits for the economy and society [44]. The economic results of smart city initiatives affect the overall economic profile of a country through the formation of new start-ups, job creation, workforce development, and the overall improvement of productivity, agencies, and the country as a whole [25]. Therefore, it is important to study the benefits, such as efficiency, efficacy, sustainability, and equality, of executing initiatives on relating to the smart economy (H4).

**H4.** *The implementation of actions relating to the economy and innovation in a smart city has a positive and significant impact on its creation, benefiting the urban ecosystem and adding value to the urban ecosystem.*

## 3. Data and Research Method

*3.1. Research Framework*

The formation of a smart city consists of three phases: phase I involves assessing the significance of smart city projects during the planning phase; phase II focuses on the implementation of planned initiatives; and phase III is the fully functional smart city or smart city 3.0. To form the conceptual framework, this paper focuses on the actions related to the smart economy and innovation during the first two aforementioned phases, which create the foundations for a smart city. It examines the correlation between the evaluation of significance during the design phase and the level of implementation of projects relating to the economy and innovation. The factors for assessing the impact, importance, and degree of implementation are (1) an attractive ecosystem for setting up new businesses (e.g., start-ups, spin-offs, spin-outs), (2) creating support mechanisms (i.e., incubators) for new businesses, (3) cooperative joint ventures, and (4) collaboration with teams from a university or research groups.

The hypotheses are based on the relationships between and impact on the selected phases and their factors (Figure 2). Evaluation is achieved through an empirical model, which includes factors for decision making, smart city strategy, and the impact of smart city factors. The currently implemented smart city strategy is investigated to determine its influence amongst phases I and II, as a mediator. Complementarily, the research takes into account the needs of the urban ecosystem that impact the planning (phase I) and implementation phase of the initiatives (phase II), relating them to enhancing the economy and innovation: (1) the city's internal operational needs, and (2) the local economy needs.

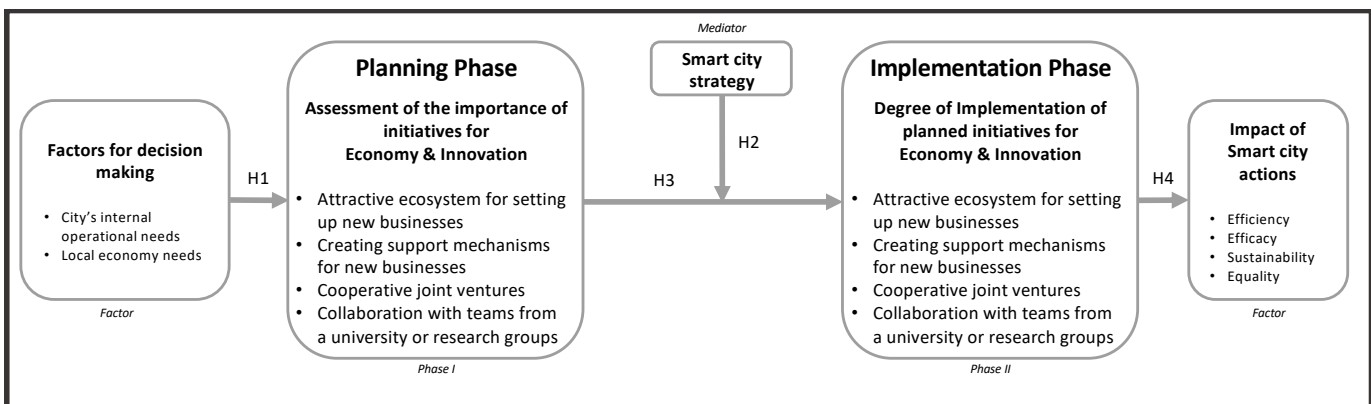

**Figure 2.** The hypothesized internal structure of the proposed research model.

*3.2. Data Specifications and Processing*

The research of this paper is part of research for a PhD focusing on smart cities in Greece and the role of the advancements in ICT in their strategy. Hence, the selected data originates from field research conducted from November 2018 to April 2019 via a structured questionnaire. It focused on three predominant elements of a Greek municipality: (i) the digital and technological characteristics, (ii) the attributes of a smart city strategy designed and executed in the last couple of years, and (iii) the features of the established collaborations with public authorities. The selected sample coincides with above 70% of the Greek municipalities and respective urban populations (252 municipalities across all 13 administrative regions of Greece (NUTS II level)). The data and their characteristics are published in the Data in Brief [45].

Based on the proposed hypotheses and research model, appropriate indicators were selected to form suitable variables to measure and assess the research model and its main components. Therefore, the crucial factors that influence the strategic planning and implementation of smart city projects may be identified. In summary, all the measurement properties of the observed indicators and their constructs, including their description, their descriptive statistics [mean and standard deviation (S.D.)], and outer variance inflation factor (VIF) (max = 5) [46], are presented in Table 1. All the indicators are measured in a Likert scale from 1 (low) to 5 (max), apart from the smart city strategy, which is from 1 (no existing strategy) to 4 (complete strategy for smart cities). All indicators are grouped according to their conceptual relevance and validated through confirmatory factor analysis (CFA). Taking into consideration the model's features, the latent variables satisfy the statistical criteria for inclusion and, thus, are classified as reflective [47].

**Table 1.** The main characteristics of the variables (latent and observed) [1].

| Construct Label | Synthesis | Depiction | Mean | S.D. | VIF [2] |
|---|---|---|---|---|---|
| Assessment of importance [P] | Evaluate the importance of initiatives for the economy and innovation which are directly designed to be executed in the municipality and refer to: | | | | |
| | [P] Attractive Ecosystem | Attractive ecosystem for setting up new businesses (e.g., start-ups, spin-offs, spin-outs) | 3.12 | 1.317 | 3.849 |
| | [P] Support mechanisms | Creating support mechanisms (i.e., incubators) for new businesses | 2.60 | 1.335 | 3.294 |
| | [P] Joint Ventures | Cooperative joint ventures | 2.96 | 1.282 | 3.143 |
| | [P] Uni/Research Teams | Collaboration with teams from a university or research groups | 2.89 | 1.277 | 2.563 |

**Table 1.** *Cont.*

| Construct Label | Synthesis | Depiction | Mean | S.D. | VIF [2] |
|---|---|---|---|---|---|
| Degree of implementation [I] | Evaluate the importance of initiatives for the economy and innovation which are being or are implemented in the municipality and related to: | | | | |
| | [I] Attractive Ecosystem | Attractive ecosystem for setting up new businesses | 2.30 | 1.099 | 2.336 |
| | [I] Support mechanisms | Collaborative formations | 1.87 | 1.075 | 2.393 |
| | [I] Joint Ventures | Creating an incubator for new businesses | 2.22 | 1.075 | 2.603 |
| | [I] Uni/Research Teams | Collaboration with university research teams | 2.27 | 1.104 | 1.894 |
| Factors for decision making | Internal business needs | Functions related to its internal business needs | 2.97 | 1.189 | 2.564 |
| | Needs of local economy | Functions related to the needs of the local economy | 2.90 | 1.193 | 2.564 |
| Impact of smart city actions | Efficiency | Efficiency (e.g., increasing the productivity of municipal employees) | 3.58 | 1.089 | 3.144 |
| | Efficacy | Efficacy (e.g., production of more projects) | 3.59 | 1.105 | 3.433 |
| | Sustainability | Sustainability (e.g., existence of economic development in the region and development of new businesses) | 3.52 | 1.116 | 2.605 |
| | Equality | Equality (e.g., access of all citizens to all services) | 3.65 | 1.119 | 2.437 |
| Smart city strategy [3] | Smart city strategy | The level of extent in which a municipality has a digital strategy and a strategy of smart cities | 1.67 | 1.097 | 1.000 |

[1] Results of the statistical test using 5000 iterations. [2] Full collinearity test of measurement model. [3] Single-item construct-mediator.

### 3.3. Research Method

Based on the conceptual model and data characteristics, the research method applied for investigating the variables' conceptual relationships is the advanced statistical technique, Structural Equation Modelling (SEM). This technique is distinguished into two methods: Covariance Based—Structural Equation Modelling (CB-SEM) and Partial Least Square—Structural Equation Modelling (PLS-SEM). A set of criteria was followed to set a valid selection process. PLS-SEM is for exploratory research and theory development (prediction), and for research models with fewer indicators, especially single-item constructs, and its measurement focuses on the total variance. In contrast, CB-SEM is a method used for explanation research and theory confirmation, while the calculations are based on common variance and are validated using the Global goodness-of-fit indices [48].

A series of mathematical equations of PLS-SEM are applied for estimating the indicators. The estimation of the latent variables' scores in the measurement (1) and structural (2) model are used for calculating the outer and inner approximation of the latent scores and weights in the measurement mode, respectively [49].

$$x_{11} * w_{11} + x_{12} * w_{12} + \ldots + x_{1m} * w_{1m} = y_1$$
$$x_{21} * w_{21} + x_{22} * w_{22} + \ldots + x_{2k} * w_{2k} = y_2$$
$$\ldots$$
$$x_{n1} * w_{n1} + x_{n2} * w_{n2} + \ldots + x_{nz} * w_{nz} = y_n \tag{1}$$

$$y_1 * b_1 + y_2 * b_2 + \ldots + y_{n-1} * b_{n-1} = y_n$$
*(all the latent variables are connected to the latent variable $y_n$)* $\tag{2}$

where, $x_{ij}$: the value of the independent indicator, $w_{ij}$: the loading of the indicator to the latent viable, $b_i$: the loading of the latent variables to the latent viable, $y_i$: the latent variable I, n: the number of the latent variables (i = 1, ..., n), and m, k, z: the number of the independent indicators per latent variable (j = 1, ..., m/k/z).

Furthermore, a set of equations are used for assessing the models presented. Each equation from (3) to (8) is used for calculating a selective indicator [50,51].

$$CA = \left(N \times \left(COV\left(X_{ij}, Y_i\right) \div N\right)\right) \div \left(\left(\sum_{i=1}^{N} Var\left(X_{ij}\right) \div N\right) + \left((N-1) \times \left(COV\left(X_{ij}, Y_i\right) \div N\right)\right)\right) \tag{3}$$

$$\rho_A := \left(\widehat{w}'\widehat{w}\right)^2 \left(\left(\widehat{w}'(S - diag(S))\widehat{w}\right) \div \left(\widehat{w}'\left(\widehat{w}\widehat{w}' - diag\left(\widehat{w}\widehat{w}'\right)\right)\widehat{w}\right)\right) \tag{4}$$

$$\mathrm{CR} = \left(\sum_{i=1}^{N} w_{ij}\right)^2 \div \left(\sum_{i=1}^{N} w_{ij}^2 + \sum_{i=1}^{N} e_{ij}\right) \tag{5}$$

$$\mathrm{AVE} = \sum_{i=1}^{N} w_{ij}^2 \div \left(\sum_{i=1}^{N} w_{ij}^2 + \sum_{i=1}^{N} Var\left(e_{ij}\right)\right) \tag{6}$$

$$HTMT_{ij} = \frac{1}{w_i w_j} \sum_{g=1}^{w_i} \sum_{h=g+1}^{w_j} X_{ig,jh} \div \left(\frac{2}{w_i(w_i-1)} \sum_{g=1}^{w_i-1} \sum_{h=g+1}^{w_i} X_{ig,ih} \times \frac{2}{w_j(w_j-1)} \sum_{g=1}^{w_j-1} \sum_{h=g+1}^{w_j} X_{jg,jh}\right)^{\frac{1}{2}} \tag{7}$$

$$\mathrm{f}^2 = \left(\mathrm{R}^2_{\text{included}} - \mathrm{R}^2_{\text{excluded}}\right) \div \left(1 - \mathrm{R}^2_{\text{included}}\right) \tag{8}$$

where, $N$: the total number of indicators, k: the number of items (dimentions), $\widehat{w}$ the estimated weight vector of the latent variable (the number of indicators of the latent variable is its dimension), $f^2$: effect size, $e_{ij}$: measurement error of the item $ij$, $Var(e_{ij})$: the variance of the error of item $ij$, and $S$: empirical covariance matrix of the latent variable's indicators.

Additionally, a set of Equations (9) and (10) are presented for calculating the total and indirect effects. These equations are calculated for all the paths (arrows) presented in the model.

$$Y_{k \rightarrow z_{indirect}} = (Y_k \times Y_m \times Y_z) + (Y_k \times Y_i \times Y_z) + (Y_k \times Y_i \times Y_{i+1} \times Y_z) \tag{9}$$
*(the $Y_i$ presents the different paths connecting the two latent variables k and z)*

$$Y_{k \rightarrow z_{total}} = Y_{k \rightarrow z_{direct}} + Y_{k \rightarrow z_{indirect}} \tag{10}$$

where $Y_j$: the effect at each connection (step of the path) of different latent variables I, and $Y_{k \rightarrow z_{indirect}}$: the indirect effect of $Y_k \rightarrow Y_z$ via the different potential paths.

Consequently, even though CB-SEM is the most widely used approach, taking into consideration the nature of the research and the characteristics of the data, the preferred method is PLS-SEM. The preferred method is applicable for identifying key constructs and estimating causal relationships, while increasing the variance between dependent constructs. The next steps include evaluating robustness, nonlinearity, endogeneity, and heterogeneity results in both the measurement (outer model) and structural models (inner model), using the software SmartPLS 4.0 for analysis [52].

## 4. Model Verification and Results

### 4.1. Verification of the Measurement Model

Initially, the measurement model is formulated to assess the reliability and validity of the relationships between the observed variables and their corresponding latent variables [53]. The loading of each construct ranged from 0.811 to 0.946 ($p < 0.001$). The selected criteria to evaluate and test the model are via the Cronbach's Alpha (CA), the Dijkstra–Henseler's rho ($\rho_A$), the Composite Reliability (CR), and the Average Variance Extracted (AVE), which evaluate the construct reliability and validity of the four reflective, latent variables (Table 2) [54].

**Table 2.** Summary of the measurement model's assessment [1].

|  | Cronbach's Alpha (CA) | Dijkstra–Henseler's Rho ($\rho_A$) | Composite Reliability (CR) | Average Variance Extracted (AVE) |
| --- | --- | --- | --- | --- |
| Assessment of importance [P] | 0.921 | 0.921 | 0.944 | 0.808 |
| Degree of implementation [I] | 0.879 | 0.885 | 0.917 | 0.735 |
| Factors for decision making | 0.877 | 0.878 | 0.942 | 0.890 |
| Impact of smart city actions | 0.910 | 0.938 | 0.936 | 0.785 |

[1] The single-item construct (smart city strategy) is not presented as its value is equal to 1.000.

All four criteria exceed the threshold values, ensuring construct reliability and convergent validity [53]. To add to the above assessment, the measurement model's discriminant validity is acceptable (higher threshold: criteria HTMT < 0.850), indicating a clear conceptual distinction for the latent variables [55] (Table 3).

**Table 3.** Discriminant validity via Pearson correlation and HTMT [1].

|  | Constructs | 1 | 2 | 3 | 4 | 5 |
| --- | --- | --- | --- | --- | --- | --- |
| 1 | Assessment of importance [P] |  | 0.564 ** | 0.389 ** | 0.250 ** | 0.204 ** |
| 2 | Degree of implementation [I] | 0.625 |  | 0.393 ** | 0.206 ** | 0.303 ** |
| 3 | Factors for decision making | 0.433 | 0.445 |  | 0.303 ** | 0.450 ** |
| 4 | Impact of smart city actions | 0.265 | 0.220 | 0.340 |  | 0.163 ** |
| 5 | Smart city strategy | 0.213 | 0.322 | 0.481 | 0.171 |  |

[1] Taking the diagonal line as reference, above are the Pearson correlation and below are the HTMT values between each construct. The constructs have a mean = 0 and S.D. = 1. ** Significant at the 0.05 level (2-tailed).

In Figure 3, the outputs for the measurement and structural model are incorporated, including loadings and the path coefficient (direct effects), respectively. The analysis through PLS-SEM provides results for an in-depth understanding of the importance of the relationships between the constructs.

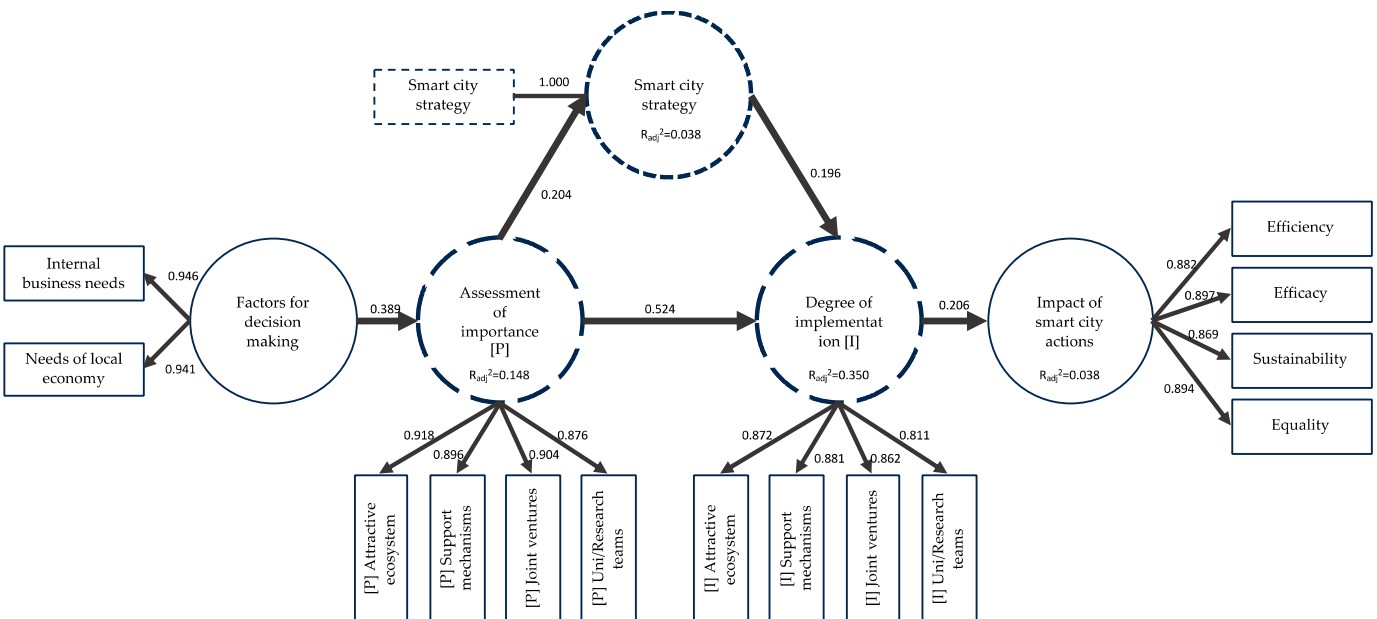

**Figure 3.** Diagram of the measurement and structural model with the loadings and the direct effects, respectively. The *p*-values are omitted to avoid clutter as they are all equal to 0.000 and the adjusted R-square are presented inside the constructs.

### 4.2. Verification of the Structural Model

In the next step, the ability of the structural model to forecast the concept is evaluated [55]. The results are presented in Table 4. Our findings show the model's ability to predict the latent constructs [54] and no significant multicollinearity problems [56].

**Table 4.** Summary of the inner model's assessment [1].

| Construct | R-Squared Adjusted | Effect Size ($f^2$) | Inner VIF [1] |
|---|---|---|---|
| | | on "Degree of Implementation" | |
| Assessment of importance [P] | 0.148 | 0.408 | 1.044 |
| Degree of implementation [I] | 0.350 | - | - |
| Factors for decision making | - | - | - |
| Impact of smart city actions | 0.038 | 0.044 | 1.000 |
| Smart city strategy | 0.038 | 0.057 | 1.044 |

[1] Full collinearity test of structural model.

Additionally, the paper evaluated the mediating effect of the smart city strategy, as a factor that support the execution of smart city projects relating to the economy and innovation. To comprehend the impact of the mediation effect, the direct and indirect effects of the paths in Table 5 are presented. Based on the analysis, the primary impact of the "*Smart city strategy*" is significant for all its connections.

**Table 5.** Summary of the direct and indirect effect of the model's correlations (*t*-values are shown in parentheses) [1].

| No. | Relationships | Direct | Indirect |
|---|---|---|---|
| 1 | Assessment of Importance → Degree of Implementation | 0.524 (10.900) *** | 0.040 (2.425) ** |
| 2 | Assessment of Importance → Impact | | 0.116 (3.267) *** |
| 3 | Assessment of Importance → Smart City Strategy | 0.204 (3.702) *** | |
| 4 | Decision Factors → Assessment of Importance | 0.389 (6.944) *** | |
| 5 | Decision Factors → Degree of Implementation | | 0.219 (5.342) *** |
| 6 | Decision Factors → Impact | | 0.045 (2.821) *** |
| 7 | Decision Factors → Smart City Strategy | | 0.080 (2.751) *** |
| 8 | Degree of Implementation → Impact | 0.206 (3.478) *** | |
| 9 | Smart City Strategy → Degree of Implementation | 0.196 (3.386) *** | |
| 10 | Smart City Strategy → Impact | | 0.040 (2.195) ** |

[1] ** Significant at the 0.05 level (2-tailed). *** Significant at the 0.01 level (2-tailed).

### 5. Discussion

All the compulsory conditions are met for the structured models, thus resulting in the investigation of the hypotheses (Table 6). Based on the analysis conducted, all the initial hypotheses are confirmed.

The findings in this paper are aligned with the scientific results presented in previous work [34,57], indicating that the existence of a smart city strategy can have a significant impact on implementing the relevant actions. By integrating into the daily routines of the city's stakeholders and exploiting smart technologies, smart city strategies can foster an environment for the social and economic development of the local community [34]. At the same time, it is important to have a plan for initiatives that are aligned with the needs of the municipality and the business ecosystem. The interaction between the different public and private actors can bring the city closer to its stakeholders. The partnerships that are developed gradually can work as mechanisms to achieve the desired outcomes and fully transform the city into a sustainable smart city [57].

**Table 6.** The status of paper's hypotheses.

| | Hypothesis | Path Coefficient (β-Value) | *t*-Value | *p*-Value | Hypothesis Status |
|---|---|---|---|---|---|
| H1 | The city's needs impact positively and significantly on the planning phase of a strategy focusing on the smart economy and innovation. | 0.398 | 6.944 | 0.000 | Confirmed |
| H2 | The assessment of planning actions in a strategy relating to initiatives focusing on economy and innovation has a significant and positive effect on the degree of implementation of relevant actions and projects. | 0.524 | 10.900 | 0.000 | Confirmed |
| H3 | The presence of a structured strategy affects indirectly, positively, and significantly the implementation phase of scheduled smart city initiatives related to the economy and innovation. | 0.040 | 2.425 | 0.015 | Confirmed |
| H4 | The implementation of actions relating to the economy and innovation in a smart city has a positive and significant impact on its creation, benefiting the urban ecosystem and adding value to the urban ecosystem. | 0.206 | 3.478 | 0.001 | Confirmed |

In this effort, the role of local government and stakeholders in planning, resources, financing, and the sustainability of actions is important. Finding and activating mechanisms, such as encouraging and engaging investors to create business initiatives and citizens to capitalize on and accept these initiatives, is a delicate balance to ensure economic and social sustainability. These roles are more important than it may seem. From that perspective, a supportive urban ecosystem offers many business perspectives and cultivates an entrepreneurial spirit, both at the individual and collective level.

Today, smart cities are presented as the solution for managing the urban phenomenon, waste, and resources. Despite their increasing growth, there is a climate of doubt about the intentions of business initiatives regarding technological development [58,59]. More specifically, although the smart city is a commonly established, flourishing market, the mechanism for the generation of value and sustainable revenues is not yet established. This discourages private sector entry without public support. However, as the majority of urban areas are evolving through public actions, the participation of the private sector with its resources is a significant factor. Therefore, municipalities need to create the appropriate settings to help the private sector to take initiative and move towards a more sustainable future.

At the same time, the role of the citizen has also changed, as someone who acquires an active position and, as a driver of innovation, contributes to the creation of value [60]. The inhabitants need to take initiative and participate in the different projects taking place in their municipality.

In potential future directions, it would be useful to include other economic factors that impact the capabilities of a municipality, like population, GDP, registered public and private entities, etc. Also, if this could be applied at a European level, it would include more diverse types of cities and could establish the foundations for a roadmap to assist local authorities to form strategies tailor-made to their need. This can be helpful for the smaller cities to attract more habitants and investment.

In summary, the dynamics of smart cities, the smart economy, and the possibilities of ICT are putting constant pressure on organizations at all levels to change. Even though the focus of an organization is profit, the new state of the urban ecosystem forces organizations to focus on value creation and public service delivery. This approach can unlock previous barriers, focusing on the needs of organizations and customers, and displaying an ability to adapt to changes.

## 6. Conclusions

The continuous implementation of ICT advancements in urban areas is of unprecedented significance. These technologies serve as essential tools for cities to progress and provide residents with essential services. However, their impact should not be viewed in isolation. Local authorities need to design strategies that not only address the needs of the local community but also cultivate an environment conducive to innovation within the private sector. Simultaneously, they should create conditions that enable citizens, as well as public and private actors, to offer value to the urban ecosystem.

The initial input taken into consideration for planning and funding initiatives within a municipality holds great significance. Actions taken by local authorities influence decisions made by the private sector, and vice versa. Therefore, it is crucial for the public sector to take on the role of a catalyst in motivating inhabitants and stakeholders to engage in activities that foster innovation and enhance the quality of life.

The findings of this research indicate the necessity for a more thorough examination of the smart economy and the influence of smart cities. Establishing a mechanism for monitoring the financial characteristics of diverse municipalities over time and assessing the initiatives undertaken by both the public and private sectors would enable a comprehensive study of a city's impact, both at the local and national levels. This, in turn, would facilitate the identification of funding requirements for public programs at national and European levels.

**Author Contributions:** Conceptualization, G.S.; methodology, G.S.; formal analysis, G.S.; writing—original draft preparation, G.S.; writing—review and editing, A.T.; supervision, A.T. All authors have read and agreed to the published version of the manuscript.

**Funding:** This research received no external funding.

**Institutional Review Board Statement:** Not applicable.

**Informed Consent Statement:** All data presented is anonymized and unified, in accordance to the guidelines provided to the participants. No participant can be identified.

**Data Availability Statement:** All the available data used for the research paper are published in Data in Brief (https://doi.org/10.1016/j.dib.2021.107716) and Mendeley Data (https://doi.org/10.17632/mpwryygc4r.1).

**Conflicts of Interest:** The authors declare no conflict of interest.

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
