# Peer review of "The Role of Economic and Innovation Initiatives in Planning a Smart City Strategy in Greece"

_sustainability, doi:10.3390/su152014842_

Round 1

Reviewer 1 Report

(1) The title of the article does not match the research content. The title is Smart City and Smart Economy, but the main focus of the entire study is on innovation and strategic issues in smart cities. It is recommended to modify the title to Smart City and Smart City Strategy

(2) The abstract is not clearly stated, and the 16th line of the abstract mentions "a tailer mate questionnaire and advanced statistical technologies". However, this questionnaire is not included in the entire article, and what is the advanced method used in the article? It is recommended to organize the abstract in a general format, such as background, research purpose, research questions, research methods, research conclusions, etc

(2) How does the title of the article reflect the absence of variables representing the economic development of Greece in Table 3? Suggest adding economic indicator variables or modifying the title.

(3) 3.3. Research method, which only briefly explains the use of PLS-SEM and CB-SEM models, why is this model used? How is it calculated? Suggest supplementing the reasons for using this model, as well as the calculation or discrimination process of the model

(4) The descriptions of Table A1 and Table 2, Table A2 and Table 3 in Appendix A on pages 8 and 10 are very confusing, and it is unclear what their relationship is? It is recommended not to use appendix format. Can we express the logical relationship between them more clearly through software screenshots or other means.

(5) In section 4.2, personal terms are used, and generally, terms such as "I", "we", and "they" are not used in articles. Please revise them carefully.

(6) The references are somewhat outdated, based on the suspicion of stacking, such as 58-65 references. It is recommended to only leave the truly referenced references and delete the listed parts. Please carefully proofread other references.

The quality of English should be properly and carefully proofread.

Author Response

We would like to thank the reviewer and the Editor for giving us the opportunity to revise and resubmit our manuscript. We, greatly, appreciate them for their complimentary comments and suggestions as we found their comments to be extremely helpful in improving our paper.

We have incorporated the suggestions made by all the reviewers, responding to each suggestion. The revisions suggested by the reviewers have been applied to the manuscript, and we have selected specific comments for discussion within this letter. In the following, we provide responses on how we specifically address each comment.

Thank you again for your consideration of our revised manuscript and we hope that you find our responses satisfactory, and that the manuscript is improved.

Reviewer 1

Comment from Reviewer 1:

The title of the article does not match the research content. The title is Smart City and Smart Economy, but the main focus of the entire study is on innovation and strategic issues in smart cities. It is recommended to modify the title to Smart City and Smart City Strategy.

Author response:

Thank you for reviewing our work and sending us your comments to improve our work. We have proceeded with changing the title to “The role of initiatives in economics and innovation in planning a smart city strategy in Greece” according to your suggestions.

Comment from Reviewer 1:

The abstract is not clearly stated, and the 16th line of the abstract mentions "a tailer mate questionnaire and advanced statistical technologies". However, this questionnaire is not included in the entire article, and what is the advanced method used in the article? It is recommended to organize the abstract in a general format, such as background, research purpose, research questions, research methods, research conclusions, etc.  

Author response:

Thank you for pointing this out. We have modified the abstract accordingly to follow your instructions. The questionnaire is not included in the paper, as it is publicly available along with the dataset in the journal “Data-in-Brief”. The reference is the following: Siokas, G., & Tsakanikas, A. (2022). Questionnaire dataset: The Greek Smart cities-municipalities dataset. Data in Brief, 40, 107716, doi.org/10.1016/j.dib.2021.107716. The questionnaire is mentioned in the paper in order for someone to take a look further into it, if it is interested.

Comment from Reviewer 1:

How does the title of the article reflect the absence of variables representing the economic development of Greece in Table 3? Suggest adding economic indicator variables or modifying the title.

Author response:

In order to address both comments (the first one and this), we have revised the title to align it more closely with the variables and the research question explored in the paper.

Comment from Reviewer 1:

3.3. Research method, which only briefly explains the use of PLS-SEM and CB-SEM models, why is this model used? How is it calculated? Suggest supplementing the reasons for using this model, as well as the calculation or discrimination process of the model.

Author response:

Thank you for your suggestion. We have restructured the "3.3 Research Method" section to include two additional paragraphs that interprets the rationale behind selecting the PLS-SEM method for our analysis, as well as the details of the model's calculation or discrimination process.

Comment from Reviewer 1:

The descriptions of Table A1 and Table 2, Table A2 and Table 3 in Appendix A on pages 8 and 10 are very confusing, and it is unclear what their relationship is? It is recommended not to use appendix format. Can we express the logical relationship between them more clearly through software screenshots or other means. 

Author response:

The results presented in Tables 2 and 3 within the main body of the text, along with Tables A1 and A2 in the appendix, represent critical findings from the analysis. These tables contain the primary outcomes from both the measurement and structural model, which we have chosen and included. Their use for evaluating robustness, nonlinearity, endogeneity, and heterogeneity results in both the measurement (outer model) and structural models (inner model) as mentioned in the section “3.3 Research method”.

Additionally, we have incorporated one schematic representation that we generated based on software screenshot for clarity and completeness. To provide additional clarification, we have provided a screenshot in Figure 1 of the software for the measurement model. This hopefully will justify our choice of creating the shapes instead of using the screenshots.

Finally, we added a small paragraph in section 4.1to explain the shapes and smoothly guide the reader at the relevant sections. The paragraph is: “In Figure 1, the outputs for the measurement and structural model are incorporated, including loadings and the path coefficient (direct effects), respectively. The analysis through the PLS-SEM provides results for an in-depth understanding of the importance of the relationships between the constructs.”.

Figure 1: Diagram of the measurement and structural model with the loadings and the direct effects, respectively. The p-values are omitted to avoid clutter as they are all equal to 0.000 and the R-square adjusted are presented inside the constructs.

Comment from Reviewer 1:

In section 4.2, personal terms are used, and generally, terms such as "I", "we", and "they" are not used in articles. Please revise them carefully.

Author response:

Thank you for bringing this to our attention. We have thoroughly reviewed the entire article with the objective of eliminating or rephrasing sentences that might include personal pronouns, namely "I," "we," (lines 335, 341) and "they" (lines 12, 217, 256). All the changes are marked in the text. This suggestion is applied to any additional text incorporated in the paper.

Comment from Reviewer 1:

The references are somewhat outdated, based on the suspicion of stacking, such as 58-65 references. It is recommended to only leave the truly referenced references and delete the listed parts. Please carefully proofread other references.

Author response:

We have reviewed all the references in the text, removing outdated and redundant ones. As a result, we have retained only the necessary and updated references.

Reviewer 2 Report

Dear Authors,

Thank you for the opportunity to read your manuscript. It's really interesting, but there are some areas that need improvement.

1. Does line 44-45 really influence economically, politically and socially? Is there no cultural influence?

2. In your manuscript, you freely use the term urban ecosystem as a concept that requires a more detailed explanation.

3. Nothing is mentioned about the application of artificial intelligence. The following sources can be used for inspiration: Mills D, Pudney S, Pevcin P, Dvorak J. Evidence-Based Public Policy Decision-Making in Smart Cities: Does Extant Theory Support Achievement of City Sustainability Objectives? Sustainability. 2022; 14(1):3. https://doi.org/10.3390/su14010003 and Vitálišová, K., Vaňová, A., Ivan, A., Hačková, I., & Borsková, K. (2023, June). Impacts of Smart Governance on Urban Development. In International Conference on Computational Science and Its Applications (pp. 547-564). Cham: Springer Nature Switzerland.

4. The purpose of the study presented in section 1.2 differs from the title and arguments presented in section 1.1 because it is initially about Greek Municipalities.

in order to exploit the smart city benefits to achieve economic development

options. maybe you need to rethink the title.

5. From the presented Figure 1, it is not very clear what kind of smart city action impact is expected, what will be the output, and why the output becomes later than the impact because usually in the classical scheme there is an output and then an impact.

6. In section 2.5, a new term urban environment appeared. need to adapt to the urban ecosystem.

7. You need to explain more about your choice of case, Why Greek municipalities and not Italian ones?

8. The findings of 322-325 are really based on evidence and not on the assumptions of the authors.

9. How was the hypothesis confirmed or not confirmed?

10. Maybe it's better to have a conclusion section because I would like to read policy implications.

All the best

Author Response

We would like to thank the reviewer and the Editor for giving us the opportunity to revise and resubmit our manuscript. We, greatly, appreciate them for their complimentary comments and suggestions as we found their comments to be extremely helpful in improving our paper.

We have incorporated the suggestions made by all the reviewers, responding to each suggestion. The revisions suggested by the reviewers have been applied to the manuscript, and we have selected specific comments for discussion within this letter. In the following, we provide responses on how we specifically address each comment.

Thank you again for your consideration of our revised manuscript and we hope that you find our responses satisfactory, and that the manuscript is improved.

Reviewer 2

Comment from Reviewer 2:

Thank you for the opportunity to read your manuscript. It's really interesting, but there are some areas that need improvement.

Author response:

Thank you for your feedback and for providing us with the opportunity to revise the text and enhance our work.

Comment from Reviewer 2:

Does line 44-45 really influence economically, politically and socially? Is there no cultural influence?

Author response:

Thank you for bringing this to our attention. We have added the term “cultural” to the text (line 43), as it represents a significant aspect in the future development of the urban environment.

Comment from Reviewer 2:

In your manuscript, you freely use the term urban ecosystem as a concept that requires a more detailed explanation.

Author response:

We have included an initial explanation of the concept “urban ecosystem” as it is perceived in the paper (lines 47-54). This description helps establish the context for the subsequent analysis in our research.

Comment from Reviewer 2:

Nothing is mentioned about the application of artificial intelligence. The following sources can be used for inspiration: Mills D, Pudney S, Pevcin P, Dvorak J. Evidence-Based Public Policy Decision-Making in Smart Cities: Does Extant Theory Support Achievement of City Sustainability Objectives? Sustainability. 2022; 14(1):3. https://doi.org/10.3390/su14010003 and Vitálišová, K., Vaňová, A., Ivan, A., Hačková, I., & Borsková, K. (2023, June). Impacts of Smart Governance on Urban Development. In International Conference on Computational Science and Its Applications (pp. 547-564). Cham: Springer Nature Switzerland.

Author response:

We read the two papers mentioned in your comment. The papers gave us the opportunity to enhance our theory of the section 2.3 and the analysis leading to the first hypothesis. They are referenced in the paper from lines 159 and lines 166.

Comment from Reviewer 2:

The purpose of the study presented in section 1.2 differs from the title and arguments presented in section 1.1 because it is initially about Greek Municipalities in order to exploit the smart city benefits to achieve economic development options. maybe you need to rethink the title. 

Author response:

We have modified the title to better correspond with the variables and the research question examined in the paper. The new title of the paper is “The role of initiatives in economics and innovation in planning a smart city strategy in Greece”. Additionally, we have made minor adjustments to the abstract and the overall structure of the paper to ensure that the paper's arguments are in harmony with its analysis.

Comment from Reviewer 2:

From the presented Figure 1, it is not very clear what kind of smart city action impact is expected, what will be the output, and why the output becomes later than the impact because usually in the classical scheme there is an output and then an impact.

Author response:

Thank you for pointing this out. For us the final output is the main goal of the city. This goal is to become a fully functional sustainable smart city. Therefore, the implementation of a strategy has an impact in its urban area than gradually leads to this transformation. We have included a relevant sentence to clarify this in lines 112-114.

Comment from Reviewer 2:

In section 2.5, a new term urban environment appeared. need to adapt to the urban ecosystem.

Author response:

Thank you bringing this to our attention. Since it aligns with our perspective, we've substituted any mention of "urban environment" with "urban concept."

Comment from Reviewer 2:

You need to explain more about your choice of case, Why Greek municipalities and not Italian ones?

Author response:

This research is part of a PhD analysis, which took place in Greece. Its primary focus is on examining the role of ICT advancements in the smart city strategies of Greek cities. We have included this pertinent clarification in the text, specifically in lines 284-285.

Comment from Reviewer 2:

The findings of 322-325 are really based on evidence and not on the assumptions of the authors.

Author response:

The results presented in that section are derived from an analysis conducted in two prior papers authored by the same individuals. These earlier papers are complementary to the current analysis and results. To enhance clarity for the reader, we have made slight modifications in the text, specifically in lines 377-378, to convey this relationship.

Comment from Reviewer 2:

How was the hypothesis confirmed or not confirmed?

Author response:

To confirm or not the hypotheses, we based our analysis on two key factors. Firstly, we assessed the validity of the measurement and structural models. Consequently, in sections 4.1 and 4.2, we presented the results of these models to establish their validity. Secondly, we presented the path coefficients (β-values) and the corresponding p-values. The former helps elucidate the nature of the relationships between various causal factors and the hypotheses' outcomes, while the latter assesses the significance of these results. All hypotheses are scrutinized through both latent and observed variables, and the model enables us to analyze their significance and relationships.

Comment from Reviewer 2:

Maybe it's better to have a conclusion section because I would like to read policy implications.

Author response:

Thank you for bringing this to our attention. We have included a conclusion section with policy implications.

Reviewer 3 Report

-Title should be more concise with the necessary information.

-Literature review should be thorough where previous research should be presented. 

-State the literature gaps.

-Present different aspects of the smart city planning model in a chart.

-Data is not so recent. 

-You should use the models of other smart cities around the world to frame the research.

-The findings in this paper are aligned with previous work presented. Discuss them also. The convergence of your result with previous studies should be presented.

-Add conclusion section, where important findings should be stated briefly.

-Add some future scope. How can more innovative features be added?

Author Response

We would like to thank the reviewer and the Editor for giving us the opportunity to revise and resubmit our manuscript. We, greatly, appreciate them for their complimentary comments and suggestions as we found their comments to be extremely helpful in improving our paper.

We have incorporated the suggestions made by all the reviewers, responding to each suggestion. The revisions suggested by the reviewers have been applied to the manuscript, and we have selected specific comments for discussion within this letter. In the following, we provide responses on how we specifically address each comment.

Thank you again for your consideration of our revised manuscript and we hope that you find our responses satisfactory, and that the manuscript is improved.

Reviewer 3

Comment from Reviewer 3:

Title should be more concise with the necessary information.

Author response:

Thank you for reviewing our work and sending us your comments to improve our work. We have proceeded with changing the title to “The role of initiatives in economics and innovation in planning a smart city strategy in Greece” according to your suggestions.

Comment from Reviewer 3:

Literature review should be thorough where previous research should be presented. 

Author response:

Thank you for bringing this to our attention. We have thoroughly reviewed the literature once more and implemented the required modifications. We have retained the most crucial literature while ensuring it remains current and relevant.

Comment from Reviewer 3:

State the literature gaps.

Author response:

Thank you for pointing this out. We have carefully reviewed the initial section of the paper and integrated identified literature gaps that were discovered during our literature analysis. For example, you can find some of these changes in lines 55-57, 69-70, and 89-91.

Comment from Reviewer 3:

Present different aspects of the smart city planning model in a chart.

Author response:

The various factors we have taken into account in our model are illustrated in Figure 2. This diagram outlines the four overarching elements that we have identified as influencing the planning and execution stages of a smart city's transition to a smart economy.

Comment from Reviewer 3:

Data is not so recent. 

Author response:

This research is part of a PhD analysis, which took place in Greece. Its primary focus is on examining the role of ICT advancements in the smart city strategies of Greek cities.

Comment from Reviewer 3:

-You should use the models of other smart cities around the world to frame the research.

Author response:

We have included a paragraph refereeing to different initiatives taken from smart cities. The paragraph is added on lines 100-108.

Comment from Reviewer 3:

The findings in this paper are aligned with previous work presented. Discuss them also. The convergence of your result with previous studies should be presented.

Author response:

We have restructured the section 4. Discussion as to include findings from previous research resented.

Comment from Reviewer 3:

Add conclusion section, where important findings should be stated briefly.

Author response:

Thank you for pointing this out. We have included the relevant section in the paper.

Comment from Reviewer 3:

Add some future scope. How can more innovative features be added?

Author response:

We have integrated an analysis regarding the future scope of the research in the paper, according to your guidelines.

Round 2

Reviewer 1 Report

(1)Regarding the comments on ""The descriptions of Table A1 and Table 2, Table A2 and Table 3 in Appendix A on pages 8 and 10 are very confusing, and it is unclear what their relationship is? " It is recommended not to use the appendix method. The author has explained some of them, but still cannot clearly explain the content. It is recommended to directly place the charts in the main text.

(2)Review comments on '3.3. Research method, which only briefly explains the use of PLS-SEM and CB-SEM models, why is this model used? How is it calculated? Suggest supplementing the reasons for using this model, as well as the calculation or discrimination process of the model."The author only replied to the method and explained that it was only divided into two paragraphs to briefly describe the use of these two methods, and multiple references were cited. It is still recommended to have clear mathematical formulas for model calculation and other content.

Reviewer 2 Report

Dear Authors,

Thank you for the updated manuscript. I think it can be accepted to the Sustainability journal. 

All the best

Reviewer 3 Report

All the revisions are ok
